# Participation in the Global Corporate Challenge^®^, a Four-Month Workplace Pedometer Program, Reduces Psychological Distress

**DOI:** 10.3390/ijerph20054514

**Published:** 2023-03-03

**Authors:** Jessica Stone, S. Fiona Barker, Danijela Gasevic, Rosanne Freak-Poli

**Affiliations:** 1School of Public Health and Preventive Medicine, Monash University, Melbourne, VIC 3004, Australia; 2Centre for Global Health, Usher Institute, The University of Edinburgh, Edinburgh EH8 9AG, UK; 3School of Clinical Sciences at Monash Health, Monash University, Melbourne, VIC 3004, Australia

**Keywords:** psychological distress, stress, physical activity, prevention, health promotion, intervention, K10, pedometer, work, occupational health, sitting, sedentary, physical activity

## Abstract

Background: Psychological distress (stress) has been linked to an increased risk of chronic diseases and is exacerbated by a range of workplace factors. Physical activity has been shown to alleviate psychological distress. Previous pedometer-based intervention evaluations have tended to focus on physical health outcomes. This study aimed to investigate the immediate and long-term changes in psychological distress in employees based in Melbourne, Australia after their participation in a four-month pedometer-based program in sedentary workplaces. Methods: At baseline, 716 adults (aged 40 ± 10 years, 40% male) employed in primarily sedentary occupations, voluntarily enrolled in the Global Corporate Challenge© (GCC©), recruited from 10 Australian workplaces to participate in the GCC^®^ Evaluation Study, completed the Kessler 10 Psychological Distress Scale (K10). Of these, 422 completed the K10 at baseline, 4 months and 12 months. Results: Psychological distress reduced after participation in a four-month workplace pedometer-based program, which was sustained eight months after the program ended. Participants achieving the program goal of 10,000 steps per day or with higher baseline psychological distress had the greatest immediate and sustained reductions in psychological distress. Demographic predictors of immediate reduced psychological distress (n = 489) was having an associate professional occupation, younger age, and being ‘widowed, separated or divorced’. Conclusions: Participation in a workplace pedometer-based program is associated with a sustained reduction in psychological distress. Low-impact physical health programs conducted in groups or teams that integrate a social component may be an avenue to improve both physical and psychological health in the workplace.

## 1. Introduction

Psychological distress represents a combination of nervousness, agitation and psychological fatigue, and is interchangeably referred to as stress [1,2]. Experiencing higher levels of psychological distress may indicate an underlying mental disorder, such as anxiety or depression [3], and has been linked to an increased risk of chronic diseases such as cardiovascular disease, arthritis and chronic obstructive respiratory disease [4]. However, there is a lack of comprehensive data collected on the incidence and prevalence of psychological distress, especially in comparison to physical health [5].

In Canada and Australia, around 10% of people report experiencing high levels of psychological distress, while 15–20% of workers across Europe and North America report experiencing psychological distress [6,7]. Psychological distress in the workplace is exacerbated by a range of workplace factors including high job demand and low job control, job strain, poor support, poor workplace relationships, low role clarity, poor organisational change management, poor organisational justice, poor environmental conditions, remote or isolated work and violent or traumatic events [8,9,10]. Work-related stress can also increase the risk of chronic disease—a study of 1,592,491 Danish workers concluded that an average of 0.25 years in women and 0.84 years in men were lost due to chronic illness associated with high job demand and low job control [11]. This can partly be explained by findings from a study of 3090 Japanese workers reporting that workers with high job demand, low job control and job strain were more likely to have pre-existing health conditions worsen as workloads and work/family conflicts arose during their employment [10]. Work-related distress is also associated with high levels of unplanned absences, sick leave, staff turnover, withdrawal, presenteeism, poor work and poor product quality [8]. Workers experiencing psychological distress at their workplace emphasise the importance of preventing and managing levels of psychological distress in working populations and identifying interventions that target and reduce psychological distress.

Physical activity has been shown to reduce psychological distress, reduce the risk of chronic disease and increase self-esteem, overall wellbeing, and health-related quality of life [12,13,14,15,16]. Mechanistically, physical activity increases the production of endorphins and neurotransmitters such as serotonin and dopamine, which boost mood and reduce feelings of stress and depression [17]. The benefit of physical activity on reducing psychological distress is irrespective of age, sex, ethnicity or having a medical condition [18,19]. A longitudinal study consisting of 33,918 observations from 17,080 individuals in the Household, Income and Labour Dynamics in Australia (HILDA) Survey over 2007, 2009 and 2011 reported that frequent participation in moderate to vigorous physical activity was associated with lower psychological distress scores [20]. A review to develop new evidence-based Australian guidelines for physical activity for adults concluded that participation in moderate to vigorous physical activity (compared to being inactive or of low levels of physical activity) was associated with a reduction in feelings of psychological distress [13]. Despite these benefits, very few adults undertake the World Health Organisation’s (WHO) recommendation of 150 min of moderate-intensity physical activity and at least 2 days of strength-based muscle training each week [21,22]. Moderate-intensity physical activity is defined as activity that is performed at 3.0–5.9 times the intensity of rest, while vigorous-intensity physical activity is performed at 6.0 or more times the intensity of rest for adults [22].

The increasingly sedentary nature of transport, leisure-time and workplaces contributes to an overall decrease in physical activity worldwide [23]. The WHO has recognised the workplace environment as an important area of action for health promotion and disease prevention [24]. In 2017, 39% of people employed in the European Union worked while sitting [25]. Attempts have been made by workplaces and research groups to reduce sedentary time and increase physical activity at work [20,26,27,28,29,30]. The Toronto Charter, reported by the International Society for Physical Activity and Health (GAPA), calls for physical activity programs that are targeted to all sections of society, including the workplace [31]. The Charter also encourages employers and academia to undertake research to provide evidence for the effectiveness of physical activity programs in work settings and to provide support for employees in workplaces to be physically active [31]. Pedometer-based interventions have been suggested as a simple method for encouraging physical activity in the general and working population [32]. 

Previous studies investigating the use of pedometers as a physical activity intervention have tended to focus on physical health outcomes rather than psychological health outcomes, and only assessed short term benefits. While there is considerable evidence that physical activity and pedometer interventions in the workplace are effective at improving health outcomes and psychological distress, further clarification is needed in future studies to address the following issues. As identified in the systematic review by Freak-Poli et al., many studies that assessed pedometer interventions and their impact on health outcomes were cross-sectional and only observed the short-term effects of the programs on health [33]. Additionally, although there is an association with lower psychological distress among people who undertake more physical activities and/or are less sedentary, these findings are not validated by changes during physical activity interventions [20,27]. 

There is also a need for physical activity interventions to assess health outcomes beyond physical health factors. Physical activity interventions primarily focus on improving physical health outcomes linked to chronic disease, but such interventions may have additional benefits. There is a need for evaluations to be expanded to include mental health outcomes as well [33]. Additionally, the Freak-Poli et al. systematic review recommends the use of longitudinal studies to follow participants over a longer period of time to demonstrate sustained long-term effects on physical and mental health outcomes after the intervention has been completed [33]. 

Furthermore, evidence shows that employees are motivated to engage in pedometer programs, as walking is a low-intensity but sustainable form of physical activity over long periods of time [33]. It is also important to note that the employees most likely to benefit from workplace low-impact walking programs are those in highly sedentary roles, such as office workers and administrative staff [33]. Women, full time workers and individuals that self-reported a healthy weight and high physical activity were more likely to engage and participate in pedometer programs, which indicates that other groups need to be targeted in future studies [33,34,35]. Such interventions may provide the opportunity to negate the negative effects associated with shift work, overtime, and high job stress, as well as improve health outcomes [36]. 

Our study aims to investigate whether participation in a four-month workplace pedometer program was associated with immediate changes (after the four-month program) and long-term changes (eight months post-program) in psychological distress. Secondly, if changes were observed, we aimed to explore factors associated with change in psychological distress. Based on previous evidence that lower psychological distress is associated with undertaking physical activity, we hypothesize that adults in sedentary occupations will have a reduction in psychological distress after participation in a group-based, low-intensity physical activity workplace program, compared to their baseline measure (pre-post design). 

## 2. Materials and Methods

This study involved secondary analysis of an existing, de-identified sample of office workers from Melbourne, Australia who were in predominantly sedentary occupations and enrolled in a group-based, pedometer workplace program. 

### 2.1. Global Corporate Challenge^®^

The Global Corporate Challenge^®^ (GCC^®^) is an annual pedometer-based, physical activity, four-month workplace health program that is conducted by a corporate organisation. The GCC^®^ is held world-wide through workplaces, which group employees into teams of seven people. In this study, participants were asked to wear the visible pedometers provided by the GCC^®^ on their hip throughout the day, with the exception of swimming and showering (it was removed during sleeping). Each participant aimed to undertake the step goal of 10,000 steps per day, which has been the historical recommended step goal to achieve adequate daily activity [12,14,37,38,39]. Each participant entered their steps into the GCC^®^ website, which was combined to generate a team step count. The team step count was displayed virtually as walking progress around a world map, with information on locations as they arrived. Teams could see their progress, as well as other teams within their company world-wide, providing a competitive edge to the program. For example, an international company can compete with the office on the other side of the world. Additionally, the team or group component of the GCC^®^ provided opportunities to get to know colleagues, external encouragement to achieve the recommended step goal, and increased collegiality among colleagues. Participants were sent weekly encouragement newsletters via email including the participant’s personal best daily step count, health tips from a nutritionist, stories from other participants, a “Dear GCC” section answering participants’ questions, housekeeping and prizes awarded by sponsors of the program. A website was used for logging daily step counts and provided access to additional health information such as the number of steps required to burn off a hamburger, communication among participants and comparing team progress. 

### 2.2. Recruitment and Participation

The GCC^®^ Evaluation Study was a prospective longitudinal observational study conducted over a 12-month period in workplaces across Melbourne (Figure 1) [12,14,37,38,39]. Participants were recruited from ten predominantly sedentary workplaces over eight weeks in April and May 2008, and were enrolled in the GCC^®^ program (Appendix A). While 716 participants completed the Kessler Psychological Distress Scale 10-item (K10) [40] at baseline, this study mainly focused on the 422 participants who completed the K10 at baseline and 4- and 12-month follow-ups. Across all variables, there was minimal missing data (Appendix B).

The GCC^®^ Evaluation Study was conducted in accordance with Monash University Human Research Ethics Approval, specifically the Standing Committee on Ethics in Research involving Humans (SCERH); Low Impact Research Project Involving Humans, project number CF08/0217-2008000125.

### 2.3. Psychological Distress

Psychological distress was measured using the 10-item Kessler Psychological Distress Scale (K10) [40]. The K10 scale is a short dimensional measure of non-specific psychological distress in the anxiety-depression spectrum [1,41]. Responses to each one of the 10 scale items were scored between 1 and 5. The final scores ranged between 10 and 50, and these were categorised as low (10–15), moderate (16–21), high (22–29) and very high (30–50) psychological distress [1,42] (further detail in Appendix C). There is significant evidence establishing the reliability and validity of the K10 across a number of diverse settings, including both international and Australian contexts, across a range of populations (Cronbach’s alpha coefficient ranges between 0.84–0.94, sensitivity 0.67–0.9 and specificity 0.74–0.81 for cut-offs below 28 [43,44,45,46,47,48,49,50,51]). K10 scores were collected at baseline, 4-months and 12-months via an online self-report survey.

### 2.4. Measures

Daily step count was used as the exposure in this study. Daily step counts were collected using pedometers (GCC^®^ brand) worn on the hip. The pedometer was manufactured by GCC^®^ and internally validated. The 10,000 daily step goal was based on previous evidence from Tudor-Locke that suggested 10,000 daily steps as indicative of active individuals [52]. Alongside the 10,000 daily step goal, we also tested the potentially new threshold of 7500 steps per day [52]. 

Covariates were assessed alongside psychological distress and step count to assess the health and psychological characteristics of participants in each psychological distress category. Potential confounders were selected a priori [53] (Clayton & Hill,1993) based on their relation with physical activity, aligned with previous papers reporting on the GCC^®^ [12,14,37,38,39]. Demographic information (age, sex, tertiary education, partner status, socio-economic status, occupation), prior participation in the GCC^®^, motivation for participation (health, to look my best, fitness, colleagues or friends and family) and behavioural measures (fruit and vegetable intake, alcohol intake, smoking status, physical activity, sitting time and takeaway dinner consumption), were collected using the core and expanded options of the WHO STEPwise approach [54] and the WHO mini-STEP [55]. Psychosocial measures of wellbeing were collected using the WHO-5 questionnaire and health-related quality of life was measured using the SF-12 [12]. Locus of control was assessed using the Duttweiler Internal Control Index [56]. 

Anthropometric measures including blood pressure, heart rate, weight, body mass index (BMI), and waist circumference were measured at baseline, 4 and 12 months. Measurements were conducted by trained staff in the morning at the employees’ workplaces using the following equipment: blood pressure (Omron IA1B Automatic blood pressure intellisense machine), height (stadiometer portable height scale code PE087and step ladder), weight (Salter electronic bathroom scales model 913 WH3R 3007 during baseline and four-month data collection and Seca digital scales model Robusta 813 during twelve-month data collection) and waist and hip measurements (Figure Finder Tape Measure Novel Products Inc. 2005 code PE024 and a mirror) [14]. 

### 2.5. Data Analysis 

The normality of K10 was assessed, with transformation undertaken if required. Baseline characteristics of study participants were stratified by categories of psychological distress and presented as mean (SD) if continuous and counts or percentages if categorical. The mean change in psychological distress in the total sample of participants that completed the K10 at all timepoints and in each psychological distress category (n = 422) was calculated using linear regression to compare changes from baseline to 4 months and baseline to 12 months. Linear regression exploratory analysis was used to investigate if other subgroups within the study had changes in psychological distress after the program. The mean change in psychological distress in participants that completed the K10 at all timepoints (n = 422) was stratified by age, sex, education, partner status, socio-economic status, occupation, motivation for participation, locus of control and step data using linear regression. Finally, linear regression analysis was used to determine predictors of immediate change in psychological distress among participants that completed the K10 at baseline and 4 months (n = 489). Factors associated with change in psychological distress were determined using univariable and multivariable (factors mutually adjusted) linear regression models. The statistical significance level was set at *p* ≤ 0.05. Data analysis was performed using Stata 16, StataCorp. 2019. Stata Statistical Software: Release 16. College Station, TX, USA: StataCorp LLC.

## 3. Results

Of the 716 participants who completed the K10 at baseline, 489 completed the K10 at baseline and 4 months, and 422 completed the K10 at baseline, four and 12 months. The K10 was slightly right-skewed, which was to be expected, as that indicated higher psychological distress (Appendix D). Hence, transformation was not required and assists with the interpretation of the findings as the K10 has prespecified categories. The smoothness of the normality of the data became disjointed with less data points at intervention completion (4 months) and long-term follow-up (12 months). Participants who only completed the K10 at baseline (n = 716) had a mean age of 40 years, 39.7% were male, and 79.9% had completed tertiary education. Participants that completed the K10 at baseline and 4 months only (n = 489) had a mean age of 41 years, 40.9% were male, and 80.6% had completed tertiary education. Of the 422 participants that completed K10 at all timepoints, had a mean age of 41, 42% were male, and 81% had completed tertiary education. Participants who remained in the study at four and 12 months (n = 422) were more likely to eat the recommended daily serving of fruits and vegetables, were more physically active and less sedentary (Appendix E). 

Among the 422 participants who completed the K10 at baseline, 4 and 12 months, participants with lower baseline psychological distress (compared to higher baseline psychological distress) were older, had lower health related motivation for participation in the program, met the recommended physical activity guidelines, consumed takeaway dinner less regularly, and had higher scores for wellbeing, the SF-12 mental health component (MCS) and internal locus of control (Table 1). 

### 3.1. Immediate and Long-Term Changes in Psychological Distress 

Psychological distress decreased by half a unit between baseline and 4 months, which was retained at the 12-month timepoint (n = 422) (Figure 2 and Appendix F). Participants with higher baseline psychological distress scores had greater reductions in psychological distress after participation in the program, while participants with low baseline psychological distress scores reported increases in psychological distress. Immediate and sustained long term reductions in K10 scores (n = 422) were observed among those who were aged 30–40, females, had completed tertiary education, were widowed, separated or divorced, associate professionals, reported that they were motivated to participate in the program due to health, to look their best, improve their fitness, or encouragement from colleagues and were more likely to meet the 10,000 daily step goal (Appendix G). 

### 3.2. Predictors of a Reduction in Psychological Distress

Univariable analysis of the 489 participants that completed the K10 at baseline and 4 months only were more likely to be younger age, being ‘widowed, separated or divorced’, being an associate professional, and achieving the goal of the program (steps average per day and meeting 10,000 steps average per day) were predictors of reductions in psychological distress after participation in the pedometer program (Table 2). The results of the multivariable analysis, when mutually adjusting for possible predictors of reduction in psychological distress, found that being employed in an associate professional occupation was the predictor with the greatest magnitude of reducing psychological distress from participation in the program (n = 453, included participants that had data for age, sex, tertiary education, socio-economic status, occupation, partner status and meeting the 10,000 steps daily goal). The associate professional occupation category, compared to the reference category of professional occupation, had the greatest magnitude of reduction in psychological distress. The immediate (4-month) and sustained (12-month) changes in psychological distress within each of these stratum are presented in Appendix H.

## 4. Discussion

Psychological distress among Australian employees in mostly sedentary workplaces was reduced after participation in the four-month workplace pedometer program, which was sustained eight months after the program ended. The reduction in psychological distress was greatest for those experiencing higher levels of stress before participating in the program. Participants achieving the goal of the program of meeting 10,000 steps average per day or with higher baseline psychological distress had the greatest immediate and sustained reductions in psychological distress. At baseline, higher psychological distress was associated with younger age, higher health related motivation for participation in the program, did not meet the recommended physical activity guidelines, consumed takeaway dinner regularly, and had lower scores for wellbeing, the SF-12 mental health component (MCS) and internal locus of control. Demographic predictors of reduced psychological distress were being an associate professional, younger age, and being ‘widowed, separated or divorced’.

### 4.1. Immediate and Long-Term Changes in Psychological Distress 

While the importance of physical activity as a factor for reducing psychological distress has been studied many times [12,13,14,15,16,17,20], there is limited evidence for this relationship during participation in a workplace pedometer program. To our knowledge, we are the second study to have assessed long term physical activity interventions that utilise pedometers in terms of psychological distress. Our findings support evidence from a prior study of 1963 Indian and Australian workplaces enrolled in the Stepathlon corporate challenge reporting a benefit in psychological distress of 0.49 (mean change) over the 100-day program period. Interestingly, both our study and the Stepathlon study are opposed to the majority of prior evidence evaluating the effectiveness of workplace physical activity interventions on psychological distress [57,58,59]. This is likely because both our study and the Stepathlon were longer programs, where the interventions were able to form a habit in the participants—a study by Lally et al., 2010 found that it takes on average two months to develop a consistent behaviour [60]. A 2019 systematic review assessing job stress during workplace exercise interventions reported that only two of eight workplace physical activity programs observed a statistically significant reduction in job stress. Another 2018 systematic review concluded that studies assessing workplace physical activity programs were of low quality due to the lack of a control group [61]. In the study by Jindo et al., the participant characteristics were similar to our study and included a lower proportion of male participants to female participants, the mean age was older (around 50 years), and participants were also mainly tertiary educated [62]. The study collected data over six months but did not find improvements in psychological distress with increased compliance in the workplace exercise program. Conversely, participants with low psychological distress at baseline had an increase in psychological distress score during and after the program. Regression to the mean [63] is expected in longitudinal studies, particularly due to the ceiling effects encountered due to the healthy cohort effect [38]. To put this into context, among the healthiest participants (the least psychologically distressed), we observed a slight increase in psychological distress. However, the magnitude of this increase would not impact psychological distress categorization greatly as small increases would shift in an individual’s score to the lower end of the moderate category or remain in the low category. Nonetheless, a bi-directional relationship between physical activity and psychological distress has been observed, where pre-existing higher levels of psychological distress are associated with decreases in physical activity [64]. Further, increases in psychological distress during participation in workplace health programs may be explained by work stressors impacting these participants during the program [10]. 

Despite the opposing evidence in the above-mentioned systematic reviews, broader literature has shown that physical activity has benefits to psychological distress. Our findings support other prior literature, such as a study by Thogersen-Ntoumani et al., which demonstrated an estimated effect size of −0.31 in enthusiasm, −0.02 in relaxation and 0.05 in nervousness, in stress-related symptoms amongst sedentary British University employees four months post-intervention (note these findings were not statistically significant) [65]. Furthermore, a study by Perales et al. assessing self-reported physical activity data from 2007, 2009 and 2011, showed estimated effects of −0.41 units on the K10 when engaging in moderate to vigorous physical activity less than once a week compared to not at all, −0.83 units for being active once or twice a week, −1.14 units for being active 3 times a week, −1.42 units for being active more than 3 times a week, and −1.79 units for being active every day [20]. This demonstrates that as individuals engaged in frequent physical activity, their psychological distress scores reduced—which aligns with the finding of our study that higher step counts were associated with higher reductions in psychological distress. 

### 4.2. Predictors of a Reduction in Psychological Distress

Our study demonstrated that people with a higher step count, higher levels of psychological distress, associate professional occupations, younger age, and being ‘widowed, separated, or divorced’ had the greatest reductions in psychological distress. Our observation that achieving 10,000 steps on average per day was associated with greater reductions of psychological distress supports the prior the Stepathlon corporate challenge study. However, the Stepathlon study also reported a benefit for participants that did not meet the 10,000 step-goal of 5.4% improvement in stress, compared to a 10.1% improvement for those meeting the goal [57]. Our magnitude of benefit was comparatively low, equating to 1.8% improvement in psychological distress among all participants and 4.5% improvement among those meeting the goal. Both our study and the Stepathlon study suggest that greater physical activity has additional benefits for psychological distress. Previous evidence also shows reductions in anxiety and depressive symptoms after moderate to intense physical activity [13]. Furthermore, recent evidence suggests a threshold of 7500 steps reduces mortality risk (hazard ratio [HR] = 0.57, [95% CI] = 0.38, 0.83), with an 8.5% mean risk reduction for every additional 1000 steps/day [58]. Findings suggest that step counts greater than 7500 daily steps only marginally reduce the magnitude of the risk (2% mean risk reduction per 1000 steps/day) [58]. However, we did not observe an association between meeting a daily step goal of 7500 steps and a reduction in psychological distress.

Our study supports prior research that identified that people with the higher levels of psychological distress received the most beneficial changes from a walking intervention [66,67]. A review of the literature has concluded that while some studies have shown higher levels of stress decreased participation in exercise and physical activity in employee populations [67], another study reported that individuals experiencing higher levels of stress engaged in higher levels of physical activity [68]. This tends to be the case for those who already engage in physical activity regularly [69] but could also be a result of life events such as new relationships, retirement, changing work conditions, income changes and personal achievements [70]. 

Being an associate professional was the strongest demographic predictor of benefiting in psychological distress from participation in the program. Job position and having increased autonomy over work has been linked to lower stress [71], however, a study in Japan has reported that professionals and managers have a higher risk of poor health compared to clerks and manual laborers [72]. At baseline, associate professionals were no more likely to be stressed than other occupations in our study, hence, physical activity interventions along with increased job autonomy could greatly benefit this group. 

The subgroups of younger age and being ‘widowed, separated or divorced’, could be targeted for low-intensity physical activity interventions to reduce stress. Among 7485 participants aged 20–64 years, higher levels of psychological distress have been observed in younger people that reported work-related stressors [71]. While we also observed a mean difference by age in psychological distress at baseline, there was only a 4-year mean difference between low and very high stress categories among participants aged 37–40 years. In our study, employees who were ‘widowed, separated or divorced’ had greater reductions in psychological distress. Evidence has shown that marriage may benefit mental health by lessening negative effects of chronic stressors, but also suggests that the changing nature of partner status can limit these effects [73]. However, we did not observe any difference in stress by partner status at baseline. Low physical activity interventions, therefore, are effective regardless of partner status, but could be a consideration in accounting for the stressors participants may have in their lives. 

There are several possible mechanisms explaining how physical activity could benefit psychological distress. Participation in a physical activity intervention over four months is likely to promote the release of endorphins and be beneficial to psychological distress [17]. We note that a bi-directional relationship may exist with pre-existing higher levels of psychological distress associated with decreases in physical activity [64]. 

### 4.3. Strengths and Limitations

The main limitation of this study is the lack of a control group, meaning a cause-and-effect relationship could not be established. This study was also undertaken during colder winter months when people are known to be less active [74]. Further, winter has also been shown to have a negative impact on psychological distress [75]. Therefore, participants could have demonstrated greater program benefits if the evaluation was repeated in the warmer months. 

Secondly, interventions and research studies typically attract participants who have positive health behaviours and therefore may perform better, known as the healthy cohort effect [38]. This may have been mitigated slightly as the GCC^®^ was available for multiple years in a row. Initial years likely recruited a healthy cohort, but over time, as more and more employees were encouraged to participate, the healthy cohort effect would reduce. Of note, psychological distress at baseline in prior participating GCC^®^ participants was no higher compared to new enrollees, however, a higher proportion of prior GCC^®^ participants completed the K10 at baseline, 4 months and 12 months (data not reported). 

Thirdly, the use of pedometers may be outdated and the pedometers are not externally validated [76]. The effect of lack of external validity is likely to be misclassification, and therefore our observed interaction between change in psychological distress and daily step count is likely to be an attenuation of any true effect. Pedometers have generally been found to be correlated with accelerometers, to have concordance with self-reported physical activity, and to have an inverse relationship with time spent sitting [52]. While we could suggest further research be undertaken utilising validated pedometers, this methodology is likely outdated. Pedometers were the device of choice for fitness programs and interventions in the early to mid-2000s. With advancements in technology, there has been a movement towards the use of accelerometers and electronic monitoring [33]. However, our findings of benefits in psychological distress are likely generalisable to studies using other technologies to monitor physical activity. Therefore, our main finding can be more generalisable to indicate that participation in a group-based, low-intensity, physical activity, walking program conducted through the workplace reduced psychological distress. Of note is our generalisation to low-intensity physical activity, as physical activity intensity can have a u-shaped association with mental health [77]. 

Fourthly, it is possible that participation in this program could have adverse consequences on psychological distress [78]. The competitive component could be experienced as encouragement or psychological distress, likely relating to the individual’s physical activity level, readiness to change, personality, and workplace politics [79]. For example, if a participant has the lowest step count in the team, they may feel pressured or shamed (rather than encouraged) to increase their daily step count. 

Further, the workplace has a number of stressors [8,9,10] and participation in a workplace health program could add to these. Despite the program being voluntary, and requiring partial payment by some employees, an employee may find participation in the program overwhelming in terms of the physical activity required or the time commitment. Therefore, the workplace health program may present another competing “job” demand. One way of coping with additional stress is psychological detachment from work, which can have positive or negative outcomes [78]. Potentially a participant may choose to increase their participation in the program as part of psychological detachment from work, thus reducing their psychological distress. Our findings demonstrate that employees with higher psychological distress received the most beneficial effects from participation in the program. Additionally, the workplace environment may provide access to people with high stressors that may not be present in other settings and therefore the effectiveness of the program might be partly attributable to the workplace setting. 

Finally, the data were collected in 2008–2009 but have been analysed through a present-day lens. In 2007–2008, around 62% of adults did not meet the recommended physical activity guidelines compared to 55% in 2017–2018 [80]. Despite the increase in meeting physical activity guidelines over time, there has been an overall decrease in manual labour occupations [81] and an increase in digital entertainment during leisure time which means that individuals are continuing to participate in highly sedentary behaviours [23]. We believe that workplaces have not changed significantly over this time and our study findings of an improvement in psychological distress from a low-impact physical activity intervention remains relevant. 

The strengths of the study include the large sample size and the use of the K10, which is used by Australian general practitioners to assess stress. Our findings are generalisable to tertiary-educated adults employed in sedentary occupations. Our findings, along with prior outcomes from the GCC^®^ Evaluation Study, fills a gap in the literature exploring pedometer-based programs and health outcomes. 

## 5. Conclusions

Among 422 predominantly sedentary employees, participation in a group-based, low-intensity, physical activity, walking program conducted in the workplace reduced psychological distress and was particularly beneficial to those with higher levels of psychological distress. Older participants that had a higher daily step count, those in associate professional occupations, and those that were ‘widowed, separated, or divorced’ had the greatest reductions in psychological distress. 

A better understanding of the relationship between physical activity and psychological distress can inform health policy. Health promotion programs can be tailored to focus interventions on overall psychological wellbeing (in addition to other health outcomes). It can be difficult to convince workplaces and employees of the value of participation in a workplace-based physical activity program; therefore, workplace policy development should reflect the need to consider the individual characteristics that affect positive health within a workplace, in order to identify and implement an appropriate intervention [82]. While improvements to workplace conditions are much needed, physical activity programs can be a complementary part of longer-term sustainable improvements in employee wellbeing. Policies concerning employee health and stress management should avoid a one-size-fits-all approach, and should focus on creating psychologically safe work environments and strengthening workplace conditions which are shown to be a major driver of employee stress. The opportunity for employees to participate in workplace group based programs that promote small positive health changes, such as the low-intensity walking program evaluated here, can be incorporated into these policies. The COVID-19 pandemic has added another dimension to workplace stress. High job demand, low job control and job strain have been shown to worsen pre-existing health conditions as workloads and work/family conflicts arose during COVID-19 lockdown and stay-at-home orders [83,84]. Low-impact physical activity interventions, such as the one evaluated in this study, can provide a solution to better physical health [38], mental wellbeing [14], and stress.

## Figures and Tables

**Figure 1 ijerph-20-04514-f001:**
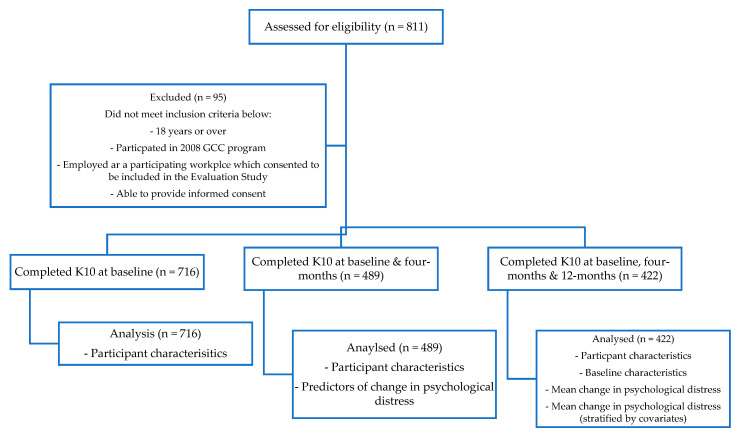
Participant recruitment.

**Figure 2 ijerph-20-04514-f002:**
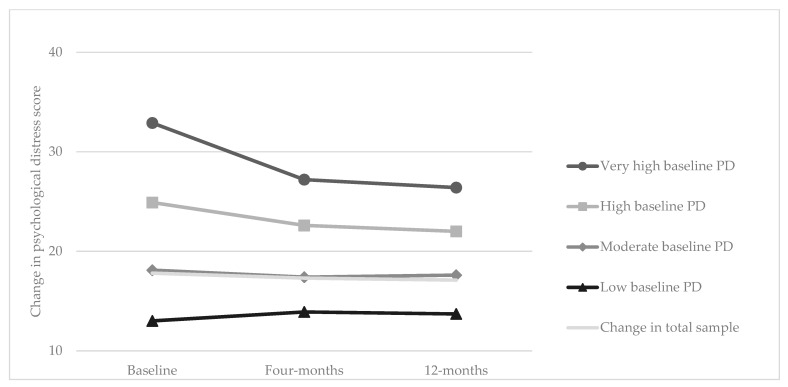
Change in psychological distress (PD) K10 score at baseline, 4 months and 12 months (n = 422) ^a^. (a K10 completed at all three timepoints).

**Table 1 ijerph-20-04514-t001:** Baseline characteristics of participants with low, moderate, high and very high levels of psychological distress (K10 scores), n = 422 ^a^.

	Psychological Distress				
N = 422	Low Mean ± SD or n (%)	Moderate Mean ± SD or n (%)	High Mean ± SD or n (%)	Very High Mean ± SD or n (%)	*p*-Value ^b^
**n**	117	215	71	19	
**DEMOGRAPHICS**					
**Age (year)**	42.6 ± 10	41.7 ± 10	39.4 ± 10.7	37.8 ± 8.8	**<0.001**
**Male**	46 (39.3%)	96 (44.7%)	29 (40.9%)	7 (36.8%)	0.366
**Completed tertiary education ^c^**	97 (82.9%)	169 (78.6%)	56 (78.9%)	18 (94.7%)	0.778
**Partner status**					
Married or de facto	86 (73.5%)	162 (75.4%)	41 (57.8%)	11 (57.9%)	0.164
Widowed, separated or divorced	9 (7.7%)	18 (8.4%)	11 (15.5%)	4 (21.1%)	
Never married	22 (18.8%)	35 (16.3%)	19 (26.8%)	4 (21.1%)	
**Socio Economic Status by residential postcode (SEIFA) ^d^**					
Most Advantaged	32 (27.4%)	49 (22.9%)	15 (21.1%)	5 (26.3%)	0.363
Advantaged	28 (23.9%)	63 (29.4%)	16 (22.5%)	3 (15.8%)	
Disadvantaged	32 (27.4%)	49 (22.9%)	18 (25.4%)	9 (47.4%)	
Most Disadvantaged	25 (21.4%)	53 (24.8%)	22 (31%)	2 (10.5%)	
**Occupation**					
Professional	49 (45%)	97 (48.3%)	29 (43.3%)	11 (64.7%)	0.768
Associate professional	24 (22%)	36 (17.9%)	16 (23.9%)	2 (11.8%)	
Manager	19 (17.4%)	42 (20.9%)	13 (19.4%)	2 (11.8%)	
Clerical or Service	17 (15.6%)	26 (12.9%)	9 (13.4%)	2 (11.8%)	
**BASELINE MEASURES**					
**Prior GCC^®^ Participation ^c^**	32 (27.4%)	49 (22.8%)	13 (18.3%)	3 (15.8%)	0.314
**Motivation for participation**					
Health ^c^	73 (62.4%)	146 (67.9%)	57 (80.3%)	13 (68.4%)	**0.006**
To look my best ^c^	67 (57.3%)	132 (61.4%)	47 (66.2%)	13 (68.4%)	0.066
Fitness ^c^	76 (65%)	144 (67%)	50 (70.4%)	14 (73.7%)	0.103
Colleagues ^c^	117 (100%)	209 (97.2%)	69 (97.2%)	18 (94.7%)	0.336
Friends or family ^c^	13 (11.1%)	23 (10.7%)	10 (14.1%)	4 (21.1%)	0.351
**BEHAVIOURAL MEASURES**					
**Fruit intake (meeting guidelines) ^c^**	36 (30.8%)	73 (34%)	27 (38%)	4 (21.1%)	0.274
**Vegetable intake (meeting guidelines) ^c^**	16 (13.7%)	38 (17.7%)	11 (15.5%)	2 (10.5%)	0.91
**Alcohol (meeting guidelines) ^c^**	57 (48.7%)	90 (41.9%)	22 (31%)	10 (52.6%)	0.676
**Non smoker ^c^**	109 (93.2%)	198 (92.1%)	68 (95.8%)	14 (73.7%)	0.284
**Physical activity (meeting guidelines) ^c^**	48 (41%)	95 (44.2%)	27 (38%)	6 (31.6%)	**0.003**
**Sitting time (hours per day)**					
Weekday	8.6 ± 3.5	8 ± 3.5	8.5 ± 3.9	8.6 ± 4.4	0.991
Weekend	5.7 ± 3	5.3 ± 2.9	4.7 ± 2.2	5.8 ± 3.8	0.437
**Takeaway Dinner**					
Once or less per month	57 (48.7%)	98 (45.6%)	32 (45.1%)	8 (42.1%)	**0.026**
About once a week	46 (39.3%)	95 (44.2%)	25 (35.2%)	10 (52.6%)	
More than once a week	14 (12%)	22 (10.2%)	14 (19.7%)	1 (5.3%)	
**PSYCHOSOCIAL MEASURES**					
**Well-being**	69.2 ± 12	63.3 ± 15.3	44.1 ± 20.2	27.8 ± 17.4	**<0.001**
**Well-being ^c^ (positive category)**	108 (92.3%)	175 (81.4%)	31 (43.7%)	3 (15.8%)	**<0.001**
**Health related quality of life (SF-12)**					
Mental health component	54.8 ± 3.6	51.4 ± 7.2	39.2 ± 11.2	31.1 ± 10.8	**<0.001**
Physical health component	50.7 ± 6.9	51 ± 7.2	51 ± 8.9	52.4 ± 7.8	0.676
**Duttweiler Internal Control Index score**	110.5 ± 10.7	106.4 ± 10.2	100.6 ± 11	97.3 ± 13.5	**<0.001**
**ANTHROPOMETRIC MEASURES**					
**Systolic blood pressure (mmHg)**	120.1 ± 12.9	117.9 ± 14.6	120.6 ± 14.9	115.5 ± 14.3	**0.053**
**Diastolic blood pressure (mmHg)**	82.1 ± 9.8	(203) 79.1 ± 10.4	(66) 79.6 ± 9.3	78 ± 10.8	**0.011**
**Heart rate (beats per minute)**	70.2 ± 11.3	(203) 68.1 ± 10.1	(66) 67.9 ± 8.5	68.1 ± 9.7	0.868
**Weight (kg)**	77.1 ± 15.5	(209) 77.6 ± 16	(68) 76.8 ± 16	(18) 81 ± 16.3	0.811
**Body mass index (kg/m^2^)**	(115) 26.8 ± 5	(209) 26.8 ± 4.6	(68) 26.7 ± 5	(18) 28 ± 5.8	0.524
**Waist circumference**	(115) 88.1 ± 12.3	(209) 88.6 ± 12.7	(68) 87.4 ± 13.1	(18) 91.9 ± 11.6	0.973
**PROCESS MEASURES**					
**STEP DATA**					
**Steps average (per day)**	11,718.5 ± 4318.3	11,839.6 ± 3368.2	(70) 11,975.9 ± 4154.3 ^e^	10,722 ± 2555.2	0.135
**Meeting 10,000 on average (per day)**					
Yes	77 (65.8%)	154 (71.6%)	46 (65.7%)	12 (63.2%)	0.168
No	40 (34.2%)	61 (28.4%)	24 (34.3%)	7 (36.8%)	
**Meeting 7500 on average (per day)**					
Yes	159 (88.3%)	141 (92.8%)	62 (88.6%)	16 (84.2%)	0.909
No	21 (11.7%)	11 (7.2%)	8 (11.4%)	3 (15.8%)	

^a^ Restricted to participants who attended and completed the K10 scale at baseline, 4-month and 12-month data collection (n = 422). ^b^ Bold highlights statistically significant results. ^c^ The reference group for this binary variable is ‘no’. The reference group data is not shown. ^d^ Socio-Economic Indexes for Areas (SEIFA) ^e^ Only 421 people who had step data that completed the K10 at all 3 timepoints. Note: percentages for some measures total greater than 100 per cent. For these measures, participants were able to select multiple responses.

**Table 2 ijerph-20-04514-t002:** Predictors of change in psychological distress (K10 score) at 4-months, n = 489 ^a^.

			Univariable Model		Multivariable Model ^b^	
Predictor Variable	n	Crude Psychological Distress Change (Units)	Psychological Distress Change B (95% CI)	*p*-Value	Psychological Distress Change B (95% CI)	*p*-Value
**Age (years)**	489	−1.4	−0.1 (−0.1, −0.01)	**0.024**	−0.02 (−0.1, 0.05)	0.549
**Sex**						
Female	200	−1.4	REFERENCE		REFERENCE	
Male	289	−1.5	0.4 (−0.7, 1.6)	0.428	−0.5 (−2.1, 1.1)	0.477
**Tertiary education**						
Not completed	95	−1.1	REFERENCE		REFERENCE	
Completed	394	−1.5	−0.4 (−2.0, 1.1)	0.539	−0.7 (−2.4, 1.0)	0.367
**Partner Status**						
Married/de facto	351	−1.4	REFERENCE		REFERENCE	
Widowed, separated or divorced	47	−3.1	−2.2 (−3.8, −0.6)	**0.012**	−2.0 (−4.3, 0.3)	0.081
Never married	91	−0.6	0.8 (−0.2, 1.7)	0.105	0.7 (−0.3, 1.7)	0.162
**Socio Economic Status by residential postcode (SEIFA)**						
Most Advantaged	29	−1.6	−0.9 (−2.0, 0.1)	0.078	−0.3 (−2.0, 1.4)	0.671
Advantaged	213	−2.2	−0.4 (−2.6, 1.7)	0.659	0.1 (−1.2, 1.4)	0.884
Disadvantaged	80	−0.9	0.2(−1.5, 1.9)	0.807	0.3 (−1.0, 1.7)	0.585
Most Disadvantaged	29	−1.0	REFERENCE		REFERENCE	
**Occupation**						
Professional	213	−1.4	REFERENCE		REFERENCE	
Associate professional	90	−2.0	−0.7 (−1.0, −0.3)	**0.004**	−1.1 (−1.8, −0.4)	**0.005**
Manager	88	−1.7	−0.7 (−1.8, 0.5)	0.226	−0.7 (−1.7, 0.3)	0.142
Clerical or Service	64	−0.3	1.1 (−0.5, 2.8)	0.162	0.7 (−0.2, 1.7)	0.123
**Steps average per day (per 10,000 steps)**	488	−1.4	−0.0001 (−0.0003, −0.00001)	**0.032**	*	
**Meeting 10,000 daily step goal**						
Yes	329	−1.8	−1.1 (−2.0, −0.2)	**0.024**	−0.6 (−1.6, 0.5)	0.272
No	159	−0.7	REFERENCE		REFERENCE	
**Meeting 7500 steps (on average) ^c^**						
Yes	440	−0.6	−1.0 (−2.6, 0.6)	0.200	−0.9 (−2.5, 0.7)	0.228
No	48	0.4	REFERENCE		REFERENCE	

^a^ Completed baseline and four-month K10. ^b^ Multivariable model (n = 453) mutually adjusted for age, sex, tertiary education, socio-economic status, occupation, partner status and meeting 10,000 steps daily goal. ^c^ Multivariable model (meeting 7500 step goal) mutually adjusted for age, sex, tertiary education, socio-economic status, occupation, partner status and meeting 7500 steps daily goal. Excluded those with data missing for age, sex, tertiary education, socio-economic status, occupation, partner status and meeting 10,000 steps daily goal variables. * Steps average per day excluded from multivariable model.

## Data Availability

The data presented in this study are available on request from the corresponding author. The data are not publicly available due to the original consent form not having a proviso for publicly available data distribution.

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
