# Peer review of "Participation in the Global Corporate Challenge®, a Four-Month Workplace Pedometer Program, Reduces Psychological Distress"

_ijerph, 2023, doi:10.3390/ijerph20054514_

Round 1
Reviewer 1 Report
Thank you for the opportunity to review this manuscript, which provided interesting information on the use of pedometers in a workplace intervention to reduce stress. There is work needed to make some of the methods more clear. Below are some comments to address:
Major comments: Address the sample numbers more clearly throughout so it is more understood how the final sample was reached and why there are different “cohorts” in the data. More detail is needed on the analyses (see my comments below).
Minor comments:
Lines 72-77: It appears there is a larger font being used here
Lines 90-91: Just a personal preference, but given you already know that changes occurred, perhaps rephrase this secondary question as “we aimed to explore factors associated with psychological distress”?
Line 93: Is this sample of 716 the same as the 422 in the abstract? I would make the numbers consistent, or explain how you got to 422 participants from 716. Similarly, you bring up n=811 in section 2.2. I would recommend making these numbers more clear throughout the manuscript
Lines 96-100: Perhaps be more specific about the covariates and how they were measured, and mention why they were included if possible. What is the rationale for using those as covariates?
Figure 1: I might recommend using the same font for the figure as the text, just for readability/consistency
Section 2.1: Where did participants wear the pedometer (e.g. wrist, hip?)? Those details matter for understanding of how the tool was used and its accuracy
Section 2.3: Can you report the data/evidence for the “good validity and reliability” for the reader? I want to know what metrics and criteria are used to determine the psychometrics are “good”
Section 2.4: Can you report the psychometrics of the pedometer? Particularly the validity of this tool to capture daily steps. Without evidence of its psychometrics, this is a limitation of the study.
Section 2.5: Did you conduct normality tests on the data? If so, please report that here. Also, what regression model did you use to select variables (e.g. stepwise, etc.)? Can you provide more information on how you entered the variables into the regression? What was the significance level cutoff to keep variables in multivariable models? What kind of regression did you run - assuming logistic but it is not clear.
For the analyses showing comparisons across the 4 K10 categories, did you use any statistical methods to account for the small sample size in the highest category of Very High (n=19)?
Section 4: Be careful with your discussion – the results seem to show reductions in stress only in those who had high stress to begin with.
Section 5: Lines 394-95, How can a better understanding of physical activity and stress inform health policy? That needs to be made more clear or the sentence can be deleted
Author Response
As detailed in the attached.

Reviewer 2 Report
This is an interesting study examining the immediate and long-term changes in psychological distress in employees after their participation in a four-month pedometer-based program in sedentary workplaces.
Authors may consider some issues highlighted below to increase clarity and improve the quality of their manuscript.
Abstract: authors may avoid using statistical indexes and related numerical results in the abstract.
The part of introduction related to the rational of the study is rather weak. Author should strength this part adding more information regarding the previous evidence related to the aim of their study, what these studies have shown, and more critically, how the present study expanded the previous ones.
The hypotheses of this study should be reported and justified.
Please provide more information to clarify the nature of the “workplace pedometer program” evaluated in this study. Participants in the study had to attend a specific walking program or just they wore the pedometers and their daily steps were accounted? The first was implied in pages 390-392 “Among 422 predominantly sedentary employees, participation in a low intensity 390 physical activity walking program …”. However, this is not clear in the description of the GCC program. Please clarify.
Why participants were organized into teams of seven but they were required to meet an individual step goal (i.e., 10,000 steps per day)? What was the purpose of organizing in groups? Has each group set a team goal? Why the comparisons were made at group level and not at individual level? Please clarify.
Can the process of comparisons between groups regarding the steps they have completed may increase participants distress? Goal setting theory suggests that personal goals and self-reference evaluation criteria are preferable for maximizing the positive effects of goal setting in performance. Why authors adopted this approach?
Authors present their results in too many Tables, Figures and Appendixes that are difficult to follow. I would suggest, authors to retain in the main text of the manuscript only the necessary results (i.e., those associated with the aims of their study) and to move the rest in supplementary files.
For example, Table 2 and Figure 2 present similar information. One of these may be presented in a supplementary file.
Please discuss the result that participants with low baseline psychological distress scores reported increases in psychological distress. Why this happened? What this result can tell us regarding the intervention program?
Author Response
As detailed in the attached

Reviewer 3 Report
The aim of the research was to determine the immediate and long-term changes in psychological distress in employees after their participation in a four-month pedometer-based program in sedentary workplaces. It has been shown that participants achieving the program goal of the 10,000 steps per day or with higher baseline psychological distress had the greatest immediate and sustained reductions in psychological distress. It has also been shown that older participants that had a higher daily step count, those in associate professional occupations, and those that were ‘widowed, separated, or divorced’ had the greatest reductions in psychological distress. In the discussion, it is worth pointing out that the importance of physical activity as a factor reducing psychological distress has been studied many times. The positive impact of various types of effort was also pointed out, not only dynamic effort such as walking, but also of a completely different nature, such as yoga or tai chi. When promoting increased physical activity as a way to reduce psychological distress, one should not forget about the need to adjust its intensity (there is a U-shaped relationship between exercise intensity and psychological stress), and type to the physical capabilities of the study participants, and even to the time of year in which the activity is to be undertaken. In addition, it should be emphasized that the time spent on physical effort is also psychological detachment from work, and increasing the ability to detach is one of the ways to cope with stress. From this point of view, it is understandable to see greater reductions in psychological distress in those subjects who achieve the program goal of the 10,000 steps per day.
Author Response
As detailed in the attached

Round 2
Reviewer 1 Report
In section 2.3, can you report the psychometrics of the K10, rather than just state that extensive work has been completed to measure reliability and validity? Perhaps list the range or an average given that multiple citations are listed here
Author Response
Thank you for ensuring our manuscript is of high quality. We trust that this is the information you are seeking (now updated in the manuscript, lines 195-199:
There is significant evidence establishing the reliability and validity of the K10 across a number of diverse settings, including both international and Australian contexts, across a range of populations (Cronbach’s alpha coefficient ranges between 0.84-0.94, sensitivity 0.67-0.9 and specificity 0.74-0.81 for cut-offs below 28 [38, 75-82])
kind regards, The Authors